# Evaluating Molecule Synthesizability via Retrosynthetic Planning and Reaction Prediction

## Abstract

A significant challenge in wet lab experiments with current drug design generative models is the trade-off between pharmacological properties and synthesizability. Molecules predicted to have highly desirable properties are often difficult to synthesize, while those that are easily synthesizable tend to exhibit less favorable properties. As a result, evaluating the synthesizability of molecules in general drug design scenarios remains a significant challenge in the field of drug discovery. The commonly used synthetic accessibility (SA) score aims to evaluate the ease of synthesizing generated molecules, but it falls short of guaranteeing that synthetic routes can actually be found. Inspired by recent advances in top-down synthetic route generation and forward reaction prediction, we propose a new, data-driven metric to evaluate molecule synthesizability. This metric leverages the synergistic duality between retrosynthetic planners and reaction predictors, both of which are trained on extensive reaction datasets. To demonstrate the efficacy of our metric, we conduct a comprehensive evaluation of round-trip scores across a range of representative molecule generative models.

## 1 Introduction

Drug design is a fundamental problem in machine learning for drug discovery. However, when these computationally predicted molecules are put to the test in wet lab experiments, a critical issue often arises: many of them prove to be unsynthesizable in practice (Parrot et al., 2023). This synthesis gap can be attributed to two primary factors. Firstly, while structurally feasible, the predicted molecules often lie far beyond the known synthetically-accessible chemical space (Ertl & Schuffenhauer, 2009). This significant departure from known chemical territory makes it extremely difficult, and often impossible, to discover feasible synthetic routes (Segler et al., 2018; Liu et al., 2023b). This synthesis challenge is underscored by numerous clinical drugs derived from natural products, which, due to their intricate structures, can only be obtained through direct extraction from natural sources rather than synthesis methods (Zheng et al., 2022). These natural products often have complex ring structures and multiple chiral centers, which makes their chemical synthesis challenging (Paterson & Anderson, 2005). Additionally, the biological processes that create these compounds are frequently not well understood, increasing the complexity of laboratory synthesis. Secondly, even when plausible reactions are identified based on literature, they may fail in practice due to the inherent complexity of chemistry (Lipinski, 2004). The sensitivity of chemical reactions is such that even minor changes in functional groups can potentially prevent a reaction from happening as anticipated.

The ability to synthesize designed molecules on a large scale is crucial for drug development. Some current methods (You et al., 2018; Gao & Coley, 2020) rely on the Synthetic Accessibility (SA) score (Ertl & Schuffenhauer, 2009) for synthesizability evaluation. This score assesses how easily a drug can be synthesized by combining fragment contributions with a complexity penalty. However, this metric has limitations as it evaluates synthesizability based on structural features and fails to account for the practical challenges involved in developing actual synthetic routes for these molecules. In other words, a high SA score does not guarantee that a feasible synthetic route for the molecule can be identified using available molecule synthesis tools (Genheden et al., 2020; Tripp et al., 2022).

To overcome the limitations of the SA score, recent works (Guo & Schwaller, 2024; Cretu et al., 2024) have employed retrosynthetic planners or AiZynthFinder (Genheden et al., 2020) to evaluate the synthesizability of generated molecules. These tools are used to find synthetic routes and assess the proportion of molecules for which routes can be found. As a result, these works rely on the search success rate for evaluating molecule synthesizability. However, this metric is overly lenient, as it fails to ensure that the proposed routes are actually capable of synthesizing the target molecules (Liu et al., 2023b). In practice, many reactions predicted by these tools may not be simulated in the wet lab, as these tools often rely on data-driven retrosynthesis models prone to predicting unrealistic or hallucinated reactions Zhong et al. (2023); Tripp et al. (2024).

To address the overly lenient evaluation metrics in previous retrosynthesis studies, where success is often defined merely by finding a "solution" without any regard to whether the solution can be executed in the wet lab (Tripp et al., 2024), FusionRetro (Liu et al., 2023b) proposes assessing whether the starting materials[1] of a predicted route of a target molecule match those in reference routes from the literature database for a target molecule. However, for new molecules generated by drug design models, reference synthetic routes are often unavailable in literature databases. This raises a critical question:

*Can data-driven retrosynthetic planners be used to evaluate the synthesizability of these molecules?*

Inspired by recent advancements that leverage forward reaction models (Sun et al., 2021) to enhance retrosynthesis algorithms and rank the top-k synthetic routes predicted by retrosynthetic planners (Schwaller et al., 2019b; Liu et al., 2024), and building on earlier work that already uses forward-model-based round-trip validation within *multi-step* retrosynthetic planning (Schwaller et al., 2020), we propose a three-stage approach that incorporates forward reaction models for evaluating molecule synthesizability to address this question. Importantly, our novelty is *not* the general idea of round-trip verification itself; rather, we formalize it into a dedicated three-stage synthesizability evaluation pipeline (applied consistently across top-$k$ multi-step routes) and use it to quantify and compare the realizability of molecules generated by different models.

Our evaluation process consists of three stages. In the first stage, we use a retrosynthetic planner to predict synthetic routes for molecules generated by drug design generative models. In the second stage, we assess the feasibility of these routes using a reaction prediction model as a simulation agent. This model attempts to reconstruct both the synthetic route and the generated molecule, starting from the predicted route's starting materials. In the third stage, we calculate the Tanimoto similarity, also called the round-trip score, between the reproduced molecule and the originally generated molecule as the synthesizability evaluation metric. Our point-wise round-trip score evaluates whether the starting materials can successfully undergo a series of reactions to produce the generated molecule.

With the round-trip score as the foundation, we develop a new benchmark to evaluate the "synthesizability" of molecules predicted by current structure-based drug design (SBDD) generative models. Our contributions can be summarized as follows:

- We recognize the limitations of the current metrics used for evaluating molecule synthesizability. Therefore, we propose the round-trip score as a metric to evaluate the synthesizability of new molecules generated by drug design models.

- We develop a new benchmark based on the round-trip score to evaluate existing generative models' ability to predict synthesizable drugs. This benchmark aims to shift the focus of the entire research community towards synthesizable drug design.

## 2 Background

### 2.1 Structure-Based Drug Design

While our newly developed benchmark is capable of evaluating a wide range of drug design models, this work specifically focuses on assessing the synthesizability of molecules generated by SBDD models. The primary goal of SBDD is to generate ligand molecules capable of binding to a specific protein binding site.

---

[1]Starting materials are defined as commercially purchasable molecules. ZINC (Sterling & Irwin, 2015) provides open-source databases of purchasable compounds, and we define the compounds listed in these databases as our starting materials.

## 2.2 Reaction Prediction (Forward)

Reaction prediction aims to determine the outcome of a chemical reaction. The task involves predicting the products $\boldsymbol{M}_p = \{\boldsymbol{m}_p^{(i)}\}_{i=1}^n \subseteq \boldsymbol{M}$ given a set of reactants $\boldsymbol{M}_r = \{\boldsymbol{m}_r^{(i)}\}_{i=1}^m \subseteq \boldsymbol{M}$, where $\boldsymbol{M}$ represents the space of all possible molecules. It's worth noting that in current public reaction datasets, such as USPTO (Lowe, 2014), only the main product is typically recorded (i.e., $n = 1$), with by-products often omitted.

## 2.3 Retrosynthesis Prediction (Backward)

Retrosynthesis aims to identify a set of reactants $\boldsymbol{M}_r = \{\boldsymbol{m}_r^{(i)}\}_{i=1}^m \subseteq \boldsymbol{M}$ capable of synthesizing a given product molecule $\boldsymbol{m}_p$ through a single chemical reaction. This process works backward from the desired product, determining the precursor molecules necessary for its synthesis.

## 2.4 Forward vs. Backward Prediction

Reaction prediction is a forward (reactants→product) mapping that is typically close to deterministic under fixed conditions, whereas retrosynthesis is a backward (product→reactants) and inherently one-to-many task that proposes multiple plausible precursor sets for the same target.

## 2.5 Retrosynthetic Planning

Retrosynthetic planning aims to predict synthetic routes for target molecules. This process works backward from the desired target, predicting potential precursor molecules that could be transformed into the target through chemical reactions. These precursors are then further decomposed into simpler, readily available starting materials. A synthetic route can be formally represented as a tuple with four elements: $\mathcal{T} = (\boldsymbol{m}_{tar}, \boldsymbol{\tau}, \mathcal{I}, \mathcal{B})$, where $\boldsymbol{m}_{tar} \in \boldsymbol{M} \backslash \mathcal{S}$ is the target molecule, $\mathcal{S} \subseteq \boldsymbol{M}$ represents the space of starting materials, $\mathcal{B} \subseteq \mathcal{S}$ denotes the specific starting materials used, $\boldsymbol{\tau}$ is the series of reactions leading to $\boldsymbol{m}_{tar}$, and $\mathcal{I} \subseteq \boldsymbol{M} \backslash \mathcal{S}$ represents the intermediates. In practice, planning is iterative: at each step a single-step retrosynthesis model proposes candidate reactant sets, and a search algorithm selects and expands the most promising nodes until all leaf nodes become purchasable starting materials.

Figure 1: For a given molecule, multiple synthetic routes can be identified within the reaction database, illustrating the diverse routes available for its synthesis.

## 2.6 Evaluation of Molecule Synthesis

Current evaluation methods for single-step reaction and retrosynthesis predictions rely on the exact match metric. This approach assesses whether the predicted results match the ground truth in the test dataset. Multiple predictions are generated, and the top-k test accuracy is reported.

Until recently, evaluation criteria for retrosynthetic planning had not reached a clear consensus, but they have now converged on a few key metrics. Among these, one of the most widely used is the success rate of finding a synthetic route within a limited number of calls for single-step retrosynthesis prediction (typically capped at 500). This metric (*search success rate*) only checks whether the planner can reach purchasable leaves within a budget, but it does not verify whether the entire proposed route is chemically executable.

To address this limitation, FusionRetro (Liu et al., 2023b) introduces a matching-based evaluation approach. This method compares the starting materials of synthetic routes predicted by retrosynthetic planners for

Figure 2: Illustration of the round-trip score calculation process. It consists of three stages: Retrosynthetic Planning, Forward Reproduction, and Similarity Computation.

target molecules with those from reference routes retrieved from literature databases. If the starting materials of a predicted route match those of any reference route, the prediction is considered accurate and successful. However, for molecules newly generated by drug design models, reference routes are often unavailable, motivating an alternative evaluation that does not depend on literature coverage. We defer the key comparison examples and qualitative analyses to Section 3.2, where we introduce the round-trip score.

## 3 Round-trip Score

### 3.1 Methodology

**Motivation.** As discussed in Sections 1 and 2, current heuristics-based metrics for evaluating molecule synthesizability, such as the SA score, fail to ensure that synthetic routes can be identified using existing data-driven molecule synthesis tools. However, these tools typically rely on the search success rate as a metric, which does not assess the feasibility of the predicted routes. Match-based metrics (Liu et al., 2023b) improve rigor when reference routes exist, but they are often inapplicable to novel molecules generated by SBDD models due to missing literature routes. Wet-lab validation would be ideal but is prohibitively expensive at scale.

To address this challenge, we note that recent forward reaction prediction models achieve top-1 accuracies exceeding 90% (Bi et al., 2021). These models can simulate reactions to verify whether the predicted synthetic routes are capable of synthesizing the target molecules. While this approach has its limitations, it is far more reliable than the search success rate and avoids the significant costs associated with wet lab experiments.

**Three-stage Evaluation Process.** Given a molecule $m$ proposed by a generative model, we first use a retrosynthetic planner to predict a synthetic route. Starting from the initial materials of this route, we then employ a reaction model to simulate wet lab experiments and reproduce the synthetic route until we reach the final molecule $m'$. Finally, we compute the Tanimoto similarity between $m$ and $m'$, which we define as the round-trip score. Figure 2 provides an illustration of the entire process.

Concretely, we decompose the pipeline into two learned components with distinct roles: (i) a *retrosynthetic planner* $g_\Theta$ that maps a target molecule to a multi-step route, i.e., $g_\Theta(m) = \mathcal{T} = (m_{tar}, \tau, \mathcal{I}, \mathcal{B})$, where $m_{tar} = m$, $\mathcal{B}$ are the leaf starting materials, and $\tau$ is an ordered sequence of reactions from $\mathcal{B}$ to $m$; and (ii) a *forward reaction predictor* $f_\Phi$ that executes (reproduces) the planned route forward: starting from $\mathcal{B}$, it predicts the product of each reaction in $\tau$ step-by-step, yielding a reproduced final molecule $m' = f_\Phi(\mathcal{T})$.

Figure 3: Comparison of evaluation metrics for retrosynthetic planning. The search success rate deems both routes successful, while a route-level verification can distinguish an incorrect intermediate step (top) from a chemically consistent route (bottom).

We then score the route by comparing the reproduced molecule $m'$ with the original target $m$ via a similarity function $Sim(\cdot, \cdot)$ (Tanimoto similarity in our implementation). The round-trip score can be mathematically expressed as follows:

$$S(m) = Sim(m, f_\Phi(g_\Theta(m))) = Sim(m, m').$$  (1)

Here $g_\Theta$ is responsible for proposing a plausible *backward* synthesis plan, while $f_\Phi$ provides an independent *forward* verification signal by simulating whether the proposed transformations can reconstruct the target.

### 3.2 Case Studies and Metric Comparison

In this subsection, we provide qualitative case studies to highlight why search success rate can be overly lenient, and how round-trip verification better reflects route executability.

Figure 3 shows two routes that both terminate at purchasable starting materials. Under search success rate, both would be counted as "successful" because the planner reaches leaf nodes within the step budget. However, the top route contains an incorrect single-step retrosynthesis prediction (highlighted), which breaks chemical consistency and would fail to reproduce the target molecule when executed forward. In contrast, the round-trip score penalizes such failures because forward reproduction yields a molecule $m'$ with low similarity to $m$. This illustrates that round-trip verification offers a stricter and more practical proxy for wet-lab feasibility when reference routes are unavailable.

## 4 Experiments

Our experiment consists of two parts. The first part focuses on assessing the reliability of the SA score, search success rate and round-trip score. The second part evaluates the synthesizability of generated molecules using the round-trip score.

### 4.1 Evaluating the Reliability of Synthesizability Metrics

Currently, retrosynthetic planners are employed to generate synthetic routes for new molecules. When the planner predicts a route, our synthesizability evaluation metrics need to differentiate between feasible and

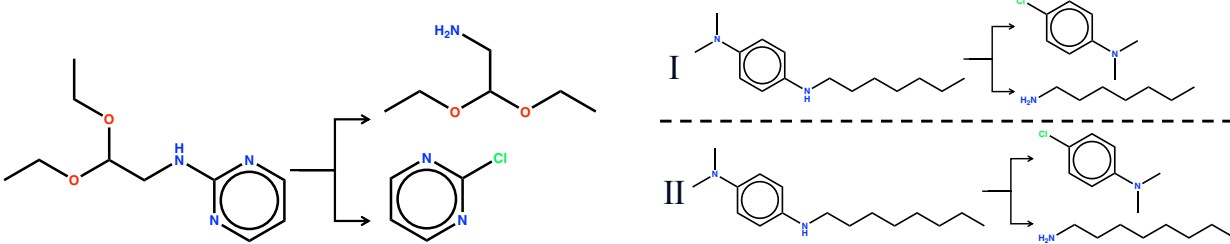

Figure 4: Examples of reactions: (a) a reaction from the CAS database confirming feasibility, and (b) a reaction predicted by the planner, differing from the CAS database reaction by one additional methyl group in the product.

infeasible routes. Therefore, we need a dataset to assess such discriminative capability of these synthesizability evaluation metrics.

**Dataset Construction.** To prepare the dataset, we first clean and deduplicate the USPTO reactions, resulting in approximately 916k reactions. These reactions are then used to construct a reaction network. Molecules with an out-degree of 0 in the network are treated as target molecules, and their corresponding synthetic routes are extracted. This process yields synthetic routes for 107,354 molecules, where the leaf nodes in the routes are starting materials. Note that some molecules can be synthesized through multiple synthetic routes in the dataset.

The dataset is divided into training, validation, and test sets with a ratio of 98%, 1%, and 1%, respectively, based on the molecules. These splits consist of 105,218, 1,068, and 1,068 data points, respectively. Each data point includes the target molecule and all its associated synthetic routes.

**Settings.** We employ the template-based Neuralsym as our retrosynthesis model, training it on reactions derived from the 105,218 data points. For predicting synthetic routes for new molecules, we leverage Neuralsym (Segler & Waller, 2017) integrated with beam search as our retrosynthetic planner. We use the Transformer (Vaswani et al., 2017) Decoder as our forward reaction prediction model, training it on about 916k reactions. All experiments in this paper are conducted using an Nvidia H100 80G GPU. We identify the primary bottleneck in search time as the retrosynthesis prediction performed by Neuralsym, which takes approximately 0.157 seconds per prediction. In contrast, our forward model, which is optimized with key-value (KV) caching and batch decoding, achieves significantly faster inference, requiring only 0.0055 seconds per reaction. Since our implementation of Neuralsym follows the original codebase from https://github.com/linminhtoo/neuralsym, it is not optimized for efficiency. As a result, retrosynthesis prediction is considerably slower than forward prediction.

**Evaluation Protocol.** We use 1,068 data points from the test set to evaluate the ability of the synthesizability evaluation metric to distinguish between feasible and infeasible routes. Please note that the number of molecules evaluated far exceeds the number of molecules used in the benchmarks for assessing retrosynthetic planning search algorithms (Chen et al., 2020). Our evaluation includes more than five times their number (189 testing molecules). Therefore, we believe the scale of our evaluation is highly convincing. For these 1,068 target molecules, we first employ the retrosynthetic planner to predict synthetic routes with a beam size of 5. During the search process, the depth of each route is limited to a maximum of 15, which is the highest depth of all routes in our dataset. While the planner can generate up to five different routes for each molecule, we only consider the route with the highest confidence score. As a result, our retrosynthetic planner successfully generates routes for 1,027 molecules, while failing to generate routes for 41 molecules.

To determine the feasibility of a predicted route, we compare it against the reference routes in the test set. If the starting materials of the predicted route match the starting materials of any reference route, the route is deemed feasible. However, it is important to note that the reference routes in the test set do not cover all possible feasible routes. For predicted routes that do not match any reference routes, we

Table 1: Performance of synthesizability evaluation metrics.

| Metric | Accuracy | Precision | Recall | F1 Score |
|---|---|---|---|---|
| Search Success Rate | - | 62.1% | - | - |
| Round-trip Score | 78.2% | 76.4% | 93.9% | 84.2% |

manually evaluate their feasibility using the Chemical Abstracts Service (CAS) SciFinder (Gabrielson, 2018) tool[2], complemented by our domain expertise, as two of our authors are Ph.D. students focusing on organic chemistry research.

Using CAS SciFinder to evaluate route feasibility, we find that all reactions in some routes are documented in the CAS database, as illustrated in Figure 4.1. Furthermore, although some reactions are not explicitly listed in the CAS database, we find that they partially match reactions documented in the database, as shown in Figure 4.1. Therefore, based on our knowledge, we think these reactions are also accurate.

Through this process, we find that for 526 molecules, the predicted routes are identified as feasible based on the reference routes in the test set. 510 of these molecules have round-trip scores of 1. For the 501 molecules whose predicted routes do not match the reference routes, feasibility is confirmed through manual evaluation and the use of CAS tools. Among these, the reactions in the predicted routes of 44 molecules are documented in the CAS database, with 36 achieving round-trip scores of 1. Furthermore, the reactions in the predicted routes of 68 molecules are considered correct based on reactions in the CAS database combined with our expertise, with 53 of these molecules attaining round-trip scores of 1. As a result, the predicted routes for 389 of the 501 molecules are considered infeasible, with 185 having round-trip scores of 1 and 204 having scores less than 1. The manual evaluation process requires five days to complete.

We evaluate the ability of synthesizability evaluation metrics to identify two types of routes: feasible and infeasible. For **feasible** routes, if the round-trip score is 1, it indicates that our forward reaction model successfully simulates the route to synthesize the target molecule, which is considered a successful identification. For **infeasible** routes, if the round-trip score is smaller than 1, it indicates that the forward reaction model fails to synthesize the target molecule by simulating the route, which is also counted as a successful identification. Based on this, we define the following terms:

- **True Positive (TP).** Correctly identified feasible routes (round-trip score = 1 for actual feasible routes): 599.

- **True Negative (TN).** Correctly identified infeasible routes (round-trip score < 1 for actual infeasible routes): 204.

- **False Positive (FP).** Incorrectly identified feasible routes (round-trip score = 1 for actual infeasible routes): 185.

- **False Negative (FN).** Incorrectly identified infeasible routes (round-trip score < 1 for actual feasible routes): 39.

Since the search success rate does not evaluate the feasibility of predicted routes, we define these terms as follows:

- **True Positive (TP).** Feasible routes correctly identified as successful search: 638.

- **False Positive (FP).** Infeasible routes incorrectly identified as successful search (routes generated but are infeasible): 389.

For the search success rate, meaningful TN and FN are absent. Therefore, we use Precision = $\frac{TP}{TP+FP}$ to compare these synthesizability evaluation metrics. Besides, we provide Accuracy = $\frac{TP+TN}{TP+TN+FP+FN}$, Recall = $\frac{TP}{TP+FN}$, and F1 Score = $2 \times \frac{\text{Precision} \times \text{Recall}}{\text{Precision} + \text{Recall}}$ for round-trip score.

---

[2]https://scifinder-n.cas.org/

Table 2: ROC-AUC scores of different synthesizability evaluation metrics.

| Metric | SYBA | GASA | SCScore | SA Score | AiZynthFinder | Round-trip Score |
|---|---|---|---|---|---|---|
| **ROC-AUC** | 0.5652 | 0.5487 | 0.4742 | 0.4511 | 0.5473 | **0.7327** |

**Results.**  Table 1 compares the precision of the round-trip score with the search success rate. The results clearly show that the round-trip score surpasses the search success rate, highlighting the forward reaction model's strength in evaluating route feasibility.

This advantage is particularly critical in minimizing false positives, as incorrectly identifying infeasible routes as feasible undermines the reliability of synthesizability evaluation metrics.

Additionally, with a recall of 93.9%, the round-trip score effectively identifies the majority of feasible routes, further demonstrating its reliability in route feasibility assessment. Moreover, as more reaction data becomes available, the accuracy of the forward reaction model is expected to improve, leading to even more reliable round-trip score evaluations. It is challenging to determine an appropriate threshold for heuristic metrics such as the SA score to reliably distinguish synthesizable molecules. Therefore, we compare the ROC-AUC scores of different evaluation metrics (SA Score (Ertl & Schuffenhauer, 2009), SYBA (Voršilák et al., 2020), SCScore (Coley et al., 2018), GASA (Yu et al., 2022a), AiZynthFinder (Genheden et al., 2020)). As shown in Table 2, our proposed metric significantly outperforms the baseline methods, demonstrating its superior discriminative capability.

### 4.2  Benchmarking Generated Molecules with Round-trip Score

In this section, we utilize the round-trip score to assess the synthesizability of molecules generated by SBDD models.

**Settings.**  To train a more powerful retrosynthesis model, we split the 107,354 molecules described in Section 4.1 into training, validation, and testing sets, allocating 107,253, 100, and 1 molecule(s) respectively. We employ the same forward reaction prediction model described in Section 4.1. During the search process, we set the beam size to 5 and limit the depth of each route to a maximum of 15. Due to computational constraints, we are unable to use a beam size of 50 as employed in retrosynthesis evaluation (Dai et al., 2019). Besides, our approach generates about 5 synthetic routes per molecule, in contrast to previous methods in retrosynthetic planning that typically produce only one route. This offers a more comprehensive evaluation compared to the search success rate metric used in earlier studies (Chen et al., 2020) for evaluating search algorithms.

**Baselines.**  For our evaluation, we employ a diverse set of state-of-the-art SBDD models, including LiGAN (Ragoza et al., 2022), AR (Luo et al., 2021), Pocket2Mol (Peng et al., 2022), FLAG (Zhang et al., 2022), TargetDiff (Guan et al., 2023a), DrugGPS (Zhang & Liu, 2023), and DecompDiff (Guan et al., 2023b). These models are trained and tested using the CrossDocked dataset (Francoeur et al., 2020), which comprises an extensive collection of 22.5 million protein-molecule structures. Our experimental setup involves randomly selecting 100,000 protein-ligand pairs from this dataset for training purposes. For testing, we draw 100 proteins from the remaining data points. To ensure a comprehensive evaluation, we randomly sample 100 molecules for each protein pocket in the test dataset, resulting in a total of 10,000 molecules. Additionally, we also verify the validity and plausibility of these molecules. After that, we employ our retrosynthetic planner to generate synthetic routes for them.

**Metrics.**  We calculate the average number of atoms in the generated molecules and the proportion that are starting materials. Besides, we calculate the percentage of molecules for which at least one of the top-k predicted synthetic routes achieves a round-trip score of 1. Please note that for molecules, which are starting materials, their round-trip scores are set to 1 without predicting any synthetic routes for them.

Table 3: The proportion of top-k predictions for each model where at least one route achieves a round-trip score of 1.

| Model | Ave. # Atoms | Rat. of Starting Materials | Top-1 | Top-2 | Top-3 | Top-4 | Top-5 |
|---|---|---|---|---|---|---|---|
| LiGAN | 21.17 | 1.66% | 2.46% | 2.60% | 2.73% | 2.79% | 2.87% |
| TargetDiff | 24.46 | 2.05% | 2.81% | 3.13% | 3.30% | 3.35% | 3.41% |
| DecompDiff | 28.34 | 0.53% | 2.84% | 3.46% | 3.78% | 4.01% | 4.05% |
| DrugGPS | 23.36 | 5.54% | 7.49% | 7.90% | 8.23% | 8.35% | 8.41% |
| AR | 17.98 | 4.67% | 7.55% | 8.32% | 8.64% | 8.87% | 9.03% |
| FLAG | 22.42 | 10.35% | 13.40% | 14.36% | 14.73% | 15.01% | 15.22% |
| Pocket2Mol | 18.53 | 14.75% | 19.45% | 20.77% | 21.45% | 21.84% | 22.05% |

**Results.**    Based on the results presented in Table 3, we can draw several conclusions. There is a significant variation in performance across different SBDD models. As the average number of atoms in the generated molecules increases, the ratio of starting materials and the top-k performance. The top-5 performance ranges from 2.87% for LiGAN to 22.05% for Pocket2Mol, indicating a substantial difference in the models' abilities to generate synthetically accessible molecules. Pocket2Mol consistently outperforms other models across all metrics, with 22.05% of its generated molecules having at least one synthetic route with a round-trip score of 1 among the top-5 predictions. The improvement in performance from top-1 to top-5 suggests that considering multiple top predictions can significantly increase the likelihood of finding feasible synthetic routes.

Notably, the performance ranking of models remains consistent across all top-k evaluations. The performance increase from top-4 to top-5 is less than 0.25%, indicating that the performance is approaching saturation. Additionally, as shown in Table 5 in Appendix B, the performance gap between models generally widens as $k$ increases, with most gaps showing an upward trend. These observations suggest that our chosen beam size of 5 is sufficient to provide an accurate ranking of each model's performance. This consistency in ranking and the approaching saturation point lend credibility to our evaluation methodology and the reliability of our comparative analysis.

The analysis of molecular properties from various generative models, as presented in Table 6, reveals that superior molecular properties do not always correlate with better synthesizability. Even for the best-performing model, a considerable portion of generated molecules still lack high-quality synthetic routes, indicating room for improvement in generating synthetically accessible molecules in SBDD tasks. These findings underscore the importance of evaluating synthetic accessibility in SBDD models and highlight the potential of using top-k predictions to identify feasible synthetic routes for generated molecules.

## 5    Related Work

**Synthesizable Drug Design.**    Existing methods for synthesizable drug design can be divided into two categories. The first category (Guo & Schwaller, 2024) directly optimizes molecular synthesizability by improving metrics such as the SA score. The second category (Bradshaw et al., 2019b; Vinkers et al., 2003; Gottipati et al., 2020; Horwood & Noutahi, 2020; Gao et al., 2022; Cretu et al., 2024; Luo et al., 2024; Koziarski et al., 2024; Seo et al., 2024) generates the final molecule step-by-step, either from building blocks or existing molecules, using defined synthesis steps and optimizing the process with reward-based methods.

**Reaction Prediction Model.**    Reaction prediction models are classified into template-based and template-free approaches. Template-based methods (Wei et al., 2016; Segler & Waller, 2017; Qian et al., 2020; Chen & Jung, 2022) predict templates to generate products. Template-free methods are more diverse, using two-stage processes to identify reaction centers and modify bonds (Jin et al., 2017), framing prediction as sequence-to-sequence or graph-to-sequence tasks (Yang et al., 2019; Schwaller et al., 2019a; Tetko et al., 2020; Irwin et al., 2022; Lu & Zhang, 2022; Zhao et al., 2022), or performing direct graph transformations on reactants (Bradshaw et al., 2019a; Do et al., 2019; Sacha et al., 2021; Bi et al., 2021).

**Retrosynthesis Model.** Retrosynthesis models are classified into three types: template-free, semi-template-based, and template-based methods, often differentiated by their use of atom mapping information. Template-free methods treat retrosynthesis as a translation (Karpov et al., 2019) or graph edit problem (Sacha et al., 2021). Template-based methods (Segler & Waller, 2017; Dai et al., 2019) utilize atom mapping to build reaction templates, framing retrosynthesis as template classification or retrieval. Semi-template-based methods (Shi et al., 2020; Somnath et al., 2021) often adopt a two-stage approach: identifying reaction centers in the product and breaking it into synthons, then transforming synthons into reactants.

**Search Algorithm.** Various search algorithms have been developed for synthetic planning, including beam search, neural A* search (Chen et al., 2020; Han et al., 2022; Xie et al., 2022), Monte Carlo Tree Search (Segler et al., 2018; Hong et al., 2021), and reinforcement learning-based methods (Yu et al., 2022b). Other notable contributions (Kishimoto et al., 2019; Heifets & Jurisica, 2012; Kim et al., 2021; Hassen et al., 2022; Li et al., 2023; Zhang et al., 2023; Liu et al., 2023a; Lee et al., 2023; Yuan et al., 2024; Tripp et al., 2024) aim to efficiently navigate the reaction space and prioritize promising synthetic routes.

## 6 Conclusion

In this work, we introduce a round-trip score to evaluate the molecule synthesizability. Extensive experiments demonstrate that our proposed metric outperforms the search success rate in assessing the feasibility of synthetic routes. We also evaluate the synthesizability of molecules generated by current SBDD models. Future works can adopt our proposed metric for designing synthesizable drugs.

Although our method is effective, it is limited by available data and cannot fully verify chemical validity—such as the correctness of reagents, reaction conditions, or expected yields.

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

# A    Reproducibility

Table 4: Hyper-parameters for the reaction model.

| Parameter | Value | | Parameter | Value |
|---|---|---|---|---|
| Max Length | 402 | | Epochs | 2000 |
| Embedding Size | 64 | | Batch Size | 128 |
| Decoder Layers | 6 | | Warmup Steps | 16000 |
| Attention Heads | 8 | | LR Factor | 20 |
| FFN Hidden | 2048 | | Scheduling | $lr = \frac{\text{lr factor} \times \min\left(1.0, \frac{0.1\,\text{num\_step}}{\text{warmup}}\right)}{\max(0.1\,\text{num\_step},\text{warmup})}$ |
| Dropout | 0.1 | | | |

Table 4 reports the hyper-parameter setting of our reaction model. For Neuralsym, we follow the setting in `https://github.com/linminhtoo/neuralsym`.

# B    Additional Experimental Results

Table 5 presents performance gaps between consecutive models.

Table 5: Performance gaps (%) between consecutive models.

| Gap Between Models | Top-1 | Top-2 | Top-3 | Top-4 | Top-5 |
|---|---|---|---|---|---|
| LiGAN to TargetDiff | 0.35 | 0.53 | 0.57 | 0.56 | 0.54 |
| TargetDiff to DecompDiff | 0.03 | 0.33 | 0.48 | 0.66 | 0.64 |
| DecompDiff to DrugGPS | 4.65 | 4.44 | 4.45 | 4.34 | 4.36 |
| DrugGPS to AR | 0.06 | 0.42 | 0.41 | 0.52 | 0.62 |
| AR to FLAG | 5.85 | 6.04 | 6.09 | 6.14 | 6.19 |
| FLAG to Pocket2Mol | 6.05 | 6.41 | 6.72 | 6.83 | 6.83 |

The properties of generated molecules presented in Table 6 are derived from two primary sources. For LiGAN, AR, Pocket2Mol, FLAG, and DrugGPS, all reported metrics (Vina Score, High Affinity, QED, SA, LogP, Lip., Sim. Train, and Div.) are extracted from the DrugGPS paper. For TargetDiff and DecompDiff, the Vina Score, High Affinity, QED, SA, and Div. metrics are sourced from the DecompDiff paper.

An analysis of Table 6 reveals a crucial insight: superior molecular properties do not necessarily translate to higher round-trip scores or search success rates. This observation underscores a critical aspect of molecular generation in drug discovery - the importance of balancing molecular quality with synthesizability. While generating high-quality molecules is essential, ensuring that these molecules are practically synthesizable is equally crucial for advancing potential drug candidates. This finding highlights the need for a holistic approach in generative models for drug discovery, one that considers both the desirable properties of molecules and their feasibility for synthesis.

## B.1    USPTO Data Cleaning

We follow a deterministic data cleaning pipeline to construct a high-quality USPTO reaction dataset from atom-mapped reaction SMILES. Given each entry with the field `ReactionSmiles` in the format `reactants > reagents > products`, we apply the following filters and transformations.

Table 6: Comparing the generated molecules' properties by different generative models. We report the means and standard deviations. The properties of the test dataset for the best results are bolded.

| Model | Vina Score (kcal/mol, ↓) | High Affinity(↑) | QED (↑) | SA (↑) | LogP | Lip. (↑) | Sim. Train (↓) | Div. (↑) |
|---|---|---|---|---|---|---|---|---|
| LiGAN | -6.03$_{\pm 1.89}$ | 0.19$_{\pm 0.26}$ | 0.37$_{\pm 0.27}$ | 0.62$_{\pm 0.20}$ | -0.02$_{\pm 2.48}$ | 4.00$_{\pm 0.92}$ | 0.41$_{\pm 0.22}$ | 0.67$_{\pm 0.15}$ |
| AR | -6.11$_{\pm 1.66}$ | 0.24$_{\pm 0.23}$ | 0.48$_{\pm 0.18}$ | 0.66$_{\pm 0.19}$ | 0.21$_{\pm 1.76}$ | 4.69$_{\pm 0.45}$ | 0.39$_{\pm 0.21}$ | 0.65$_{\pm 0.13}$ |
| Pocket2Mol | -6.87$_{\pm 2.19}$ | 0.41$_{\pm 0.23}$ | 0.52$_{\pm 0.24}$ | 0.73$_{\pm 0.21}$ | 0.83$_{\pm 2.17}$ | 4.89$_{\pm 0.22}$ | 0.36$_{\pm 0.19}$ | 0.70$_{\pm 0.17}$ |
| TargetDiff | -5.47 | 0.58 | 0.48 | 0.58 | - | - | - | 0.72 |
| FLAG | -6.96$_{\pm 1.92}$ | 0.45$_{\pm 0.22}$ | 0.55$_{\pm 0.20}$ | 0.74$_{\pm 0.19}$ | 0.75$_{\pm 2.09}$ | 4.90$_{\pm 0.14}$ | 0.39$_{\pm 0.18}$ | **0.70$_{\pm 0.18}$** |
| DecompDiff | -5.67 | **0.64** | 0.45 | 0.61 | - | - | - | 0.68 |
| DrugGPS | **-7.28$_{\pm 2.14}$** | 0.57$_{\pm 0.23}$ | **0.61$_{\pm 0.22}$** | **0.74$_{\pm 0.18}$** | 0.91$_{\pm 2.15}$ | **4.92$_{\pm 0.12}$** | **0.36$_{\pm 0.21}$** | 0.68$_{\pm 0.15}$ |

**(1) Canonical parsing and removal of auxiliary annotations.** For each reaction SMILES, we first remove any auxiliary annotations appended after whitespace (e.g., `|f:...|`) by taking the first whitespace-separated token for both the reactant and product strings. We then split reactants and products by "." into individual molecules.

**(2) Single-product selection.** We support two modes: `single` and `multi`. In this work we use `single` mode and discard reactions whose product side contains multiple disconnected products (i.e., "." appears in the product string). This matches common reaction prediction benchmarks that focus on the major product.

**(3) Validity check via RDKit SMILES/InChI round-trip.** We require that every molecule on the reactant and product sides can be parsed by RDKit (`MolFromSmiles`), and that the molecule remains valid under a SMILES→InChI→Mol conversion (`MolFromInchi(MolToInchi(...))`). Reactions failing this parsing/round-trip check are removed as invalid.

**(4) Atom-mapping consistency (1:1 mapping).** To ensure mapping correctness, we extract all atom-mapping indices from the reactant and product sides using the pattern `:(\d+)]`. We keep a reaction only if the sorted list of mapping indices on the reactant side exactly matches that on the product side. This removes reactions with missing, duplicated, or inconsistent atom-mapping.

**(5) Product-level filtering for complete atom mapping.** For each (single) product molecule, we further require that *all* atoms have an explicit mapping number (`molAtomMapNumber`). Products with missing atom-mapping annotations are discarded.

**(6) Remove trivial identity reactions.** We discard reactions where any reactant molecule is identical to the product molecule, determined by exact InChI equality. This filter removes degenerate cases such as unchanged reagents or copied species.

**(7) Remove non-contributing reactants and reagents.** We remove reactants that do not contribute atoms to the target product. Concretely, we collect the set of mapping indices appearing in the product, and keep a reactant only if it shares at least one mapping index with the product. Reactants whose atom-mapping list is `[0]` (i.e., unmapped species) are treated as reagents and removed. We additionally deduplicate reactants by InChI to avoid repeated molecules.

**(8) Construct cleaned reaction SMILES and deduplicate reactions.** For each remaining product, we construct a cleaned reaction string `(filtered_reactants)≫ (product)` using canonical RDKit SMILES. We then deduplicate the dataset by exact cleaned reaction SMILES to remove duplicates across patents, and finally save the cleaned reactions as `raw.csv` with fields `id` (PatentNumber) and `reactants≫production`.

This procedure yields a cleaned set of atom-mapped reactions with valid structures, consistent mappings, non-trivial transformations, and reactants restricted to those that contribute to the product.

