# OpenReview forum: "Evaluating Molecule Synthesizability via Retrosynthetic Planning and Reaction Prediction"
_TMLR — Rejected by TMLR_

### Review · Reviewer_MFCi · 2025-11-28

**Summary Of Contributions:**

**Contributions**

The paper proposes a new round-trip score for evaluating molecule synthesizability, defined as the Tanimoto similarity between a generated molecule and the product obtained after running a retrosynthetic planner followed by a forward reaction predictor starting from purchasable starting materials. The authors build a large reaction-network–based dataset from USPTO, annotate route feasibility using literature routes plus extensive CAS SciFinder checks, and show that the round-trip score better distinguishes feasible from infeasible synthetic routes than commonly used heuristics and search-success based metrics. They then use this metric to benchmark multiple structure-based drug design (SBDD) generative models, reporting the proportion of generated ligands for which at least one of the top‑k synthetic routes achieves a perfect round-trip score.

**Key strengths:**
1. Addresses a critical gap in AI-driven drug discovery by proposing a more rigorous synthesizability metric.
2. The benchmarking results provide a useful "reality check" for the field, showing that current SOTA models are likely overfitting to structural properties at the expense of realizability.

**Key weaknesses:**
1. The metric is tightly coupled to one particular retrosynthesis model, forward model, and reaction corpus, so its robustness across tools and chemical spaces is not fully established.
2. There is little deeper analysis of failure modes (e.g., why many “good” molecules still lack feasible routes under this pipeline).​
3. The feasibility labels rely partly on authors’ expert judgment plus CAS lookup, but there is limited analysis of labeling criteria, consistency, or potential biases.

**Audience:**

Yes

**Audience Explanation:**

AI for Science, and specifically generative chemistry/drug discovery, is a rapidly growing subfield within ML.

- Methodological Interest: Researchers working on validation loops (cycle consistency) in other domains (e.g., code generation, theorem proving) will find the "Round-trip" methodology relevant.

- Practical Application: Practitioners using generative models for drug discovery need better metrics than QED/SA score to filter candidates. This paper offers a concrete tool for that purpose.

**Broader Impact Concerns:**

The paper discusses a metric for evaluating drug design models. While generative chemistry has dual-use potential, this paper focuses on evaluation and synthesizability assessment, which is a generic safety and efficacy tool.

**Claims And Evidence:**

Yes

**Claims Explanation:**

Overall, the main empirical claims about the superiority of the round-trip score over heuristic synthesizability measures and search-success rate are supported by the experiments on the constructed USPTO-based dataset.
- The authors report precision/recall, accuracy, and ROC–AUC against feasibility labels derived from literature routes plus CAS-based manual checks, and the proposed metric substantially outperforms SA-like scores and AiZynthFinder-based success rate under these definitions.
- The benchmarking of SBDD models is also supported by clear protocols: the models, dataset (CrossDocked), sampling strategy, and the top‑k round-trip score statistics are described, and the differences across models are substantial and consistent across k.

**Requested Changes:**

**Critical Changes:**
1. Analyze failure modes of the round-trip score. The paper reports counts of true/false positives and negatives but provides limited qualitative analysis of cases where the round-trip score disagrees with the feasibility labels (e.g., round-trip score =1for routes deemed infeasible, or <1 for feasible routes); a small but systematic error analysis would help readers understand when the metric is likely to fail.​

2. Discuss dependence on specific retrosynthesis and forward models. Since the metric composes a particular Neuralsym-based planner with a specific Transformer forward model trained on USPTO reactions, it would be helpful to either (i) provide an ablation or sensitivity discussion on alternative models / training data, or (ii) more explicitly discuss how model biases and coverage limitations might shape the round-trip score and its conclusions.

**Minor Changes:**
1. Provide more complete runtime and scalability characterization. The paper notes that retrosynthesis is the main bottleneck and gives per-call timings, but it would be useful to summarize approximate wall-clock costs for the main experiments (route-feasibility study and SBDD benchmark) and discuss practical trade-offs relative to lighter heuristics.

2. Deepen analysis of SBDD benchmark results. The SBDD study shows that synthesizability varies significantly across models and is not trivially aligned with standard property metrics; additional analysis connecting specific model design choices or molecular property distributions to round-trip performance, even qualitatively, would increase the insight value of this section.

---

### Review · Reviewer_V2mD · 2025-12-05

**Summary Of Contributions:**

1. This paper proposes a round-trip score, a new metric of molecular syntheticability that is closer to wet lab experiments.
2. This paper curated a large-scale, high-quality dataset to evaluate synthetic feasibility metrics, offering comprehensive and reliable data for assessment.

**Audience:**

Yes

**Audience Explanation:**

Synthetic feasibility is crucial for AI-driven drug design and related fields. The proposed metric has the potential to enhance existing approaches to synthesizability evaluation. Additionally, the large-scale dataset used in this paper is valuable for machine learning research, as it can support further exploration of synthetic feasibility standards.

**Broader Impact Concerns:**

This paper does not raise any significant ethical concerns or broader impact issues related to its work.

**Claims And Evidence:**

No

**Claims Explanation:**

1. **Overstated Novelty Contradicts Prior Art**:
The paper repeatedly claims the "round-trip score" is a "novel, data-driven metric," but this is not supported by evidence. The core concept of round-trip accuracy for synthesizability validation was clearly established in prior work (e.g., Schwaller et al. 2019). By failing to properly contextualize its contribution against these foundational studies, the paper’s primary novelty claim is factually inaccurate.
2. **Lack of Rigor and Validation in Dataset Construction**: Despite the large dataset size, the experimental foundation is compromised by poor design and unvalidated labels. First, the test set constitutes only 1% of the data, which raises concerns about statistical power. Second, the manually curated labels for "feasible/infeasible" routes lack a critical sanity check for inter-rater reliability or external validity. A basic validation step, such as conducting a second round of blind annotation by a different expert, was omitted.
3. **Outdated and Incomplete Experimental Comparisons**: The evidence for the metric’s superiority is weakened by the use of outdated baselines. The paper primarily compares against heuristic metrics (e.g., SA score) and basic search success rates, but fails to include contemporary synthesizability assessment methods (e.g., SynFlowNet). This incomplete benchmarking prevents a convincing demonstration that the proposed metric offers a meaningful advance over the current state of the art.

**Requested Changes:**

1. The presentation of the paper could be improved. Section 2 (Background) is overly lengthy and should be condensed, with key examples placed in Section 3 (Round-trip score). A new subsection could be added to present case studies.
2. The explanation of the round-trip score in Section 3 is not sufficiently clear. The roles of $f_{\Phi}$ and $g_{\Theta}$ should be presented more clearly.
3. The details of the USPTO data cleaning process could be added to the appendix. Similarly, the manual validation of synthetic routes in the test set could also be presented in the appendix.
4. Further validation of the expert-annotated test set can be performed, such as through a second check by another expert.

---

### Review · Reviewer_ETi1 · 2025-12-31

**Summary Of Contributions:**

This submission proposes a synthesizability metric (“round-trip score”) that (1) uses a retrosynthetic planner to propose a multi-step route for a target molecule, (2) uses a forward reaction predictor to “execute” that planned route starting from the proposed starting materials, and (3) measures similarity (implemented as Tanimoto similarity; used largely as an exact-match check at 1.0) between the reproduced final product and the original target as the synthesizability score. The key claim is that this forward “verification” is a stricter and more realistic proxy than (a) heuristic SA-like scores and (b) retrosynthesis search success rate, which only checks whether a route reaches purchasable leaves.

Strengths:
1. The work is well motivated. Retrosynthesis solve rates are not perfect, and forward validation is a reasonable check.
2. The authors engaged in substantial manual validation using SciFinder to evaluate the metric’s performance.
3. The paper includes benchmarking across several SBDD models.

Weaknesses:
1. I believe the central methodological idea - round-trip accuracy - was explored rather extensively previously (e.g., Schwaller et al.), and the manuscript contains some mischaracterizations of related work.
2. The paper frames the forward model as a “substitute for wet lab experiments,” but the forward model is trained on the same data distribution as the retrosynthesis planner. This approximation may lead to false agreement. In addition, there are further methodological concerns (see below).
3. The scholarly quality falls below typical TMLR standards. While the concept is easy to follow, the presentation is confusing at times (e.g., “additional 501…” is not operationalized clearly enough for reproducibility). In other places, the manuscript spends disproportionate space on relatively standard definitions (e.g., nearly half a page on true/false positives/negatives).

**Audience:**

Yes

**Audience Explanation:**

The intersection of generative AI and drug discovery (and, more broadly, chemistry) is a major topic of interest for the TMLR audience. The “synthesis gap”- where models generate valid but difficult-to-synthesize molecules - is a well-known bottleneck, so route-level evaluation ideas are likely to be of broad interest.

**Claims And Evidence:**

No

**Claims Explanation:**

1. The round-trip metric is not novel. The paper repeatedly claims novelty for the “round-trip score” (e.g., Abstract: “a new, data-driven metric”; Section 1: “novel metric”), which I believe is factually incorrect. For example, Schwaller et al. (NeurIPS 2019 workshop) introduces round-trip accuracy, and related ideas also appear in Schwaller et al. (Chem. Sci. 2020) and LocalRetro (JACS Au 2021). A comprehensive work was also preprinted this year (Sadowski, arXiv:2510.10645). These papers may use different models and settings, but the manuscript cannot ignore prior efforts on round-trip-style validation.
2. Relatedly, cited papers such as SynFlowNet/RGFN/SynCoGen do not merely use retrosynthesis planners to evaluate molecules; rather, they directly generate synthesizable pathways (bottom-up assembly) and then use retrosynthesis to check feasibility. In that sense, these works already incorporate a “generate + verify” approach, and they do so using more independent components than models trained on the same dataset. Similarly, RAScore/RetroGNN approximate synthesizability via proxies and then use retrosynthesis to check selected molecules, which are also forms of proxy/circular validation.
3. Both the retrosynthesis planner and the forward model are trained on similar USPTO-derived data. It is possible that a “successful” round-trip indicates the two models have learned the same biases or errors in the training data. This could explain why there is still a substantial false-positive rate (reported as ~24% in the manuscript). More generally, the current evaluation does not appear to robustly disentangle truly unsynthesizable molecules from cases where the search simply failed to find an available route. Lastly, I do not understand why Tanimoto similarity is needed here: if the goal is to check identity, canonicalized SMILES (with clear stereochemistry handling) seems simpler, and the current choice reads as an unnecessary complication.
4. Lastly, a significant part of the work is due to manual validation of certain routes. However, I personally evaluated some routes that the experts found to be False (in supplemental information), and I deem MANY of them to be feasible. A short list of examples: Route 894 (amide coupling with chloroacetyl chloride and fluorobenzylpiperdinol should work, as secondary amines are a lot more nucleophilic compared to alcohols), Route 120 (Sonogashira coupling with a alkynoic acid and aryl halide, the carboxylic acid does not necessarily inhibit reactions), Route 407 (Williamson ether synthesis of a bromoether and a thiophene, appears very straightforward), Route 41 (carbamoylation of a secondary amine, seems quite possible), Route 21 (SnAr/N-arylation with a heteroaryl chloride and a secondary amide, this is a little tricky as amide nitrogens are weak nucleophiles, but I fully expect it to work with the right catalyst.)

**Requested Changes:**

1. The authors must **properly** discuss all existing round-trip scores (Schwaller et al.) as well as related "circular-validation" works such as SynFlowNet. The paper needs to explain how this work differs from existing scores and compare against them. The novelty claims require significant revision, and the actual contribution must be clarified.

2. The authors must address the fact that the planner and forward model are trained on the same dataset (USPTO). This should include a reaction-level split or ablation where the forward model is trained without reactions present in test routes (or without any reactions whose products match test targets), with re-reported Table 1/2-style results.

3. The authors should discuss known limitations of round-trip metrics (e.g., not verifying full chemical validity such as reagents/conditions/yields), as noted in prior work (e.g., Digital Discovery 2024, 558). This also requires toning down any claims that the method can substitute for “wet lab experiments.”

Less critical:
1. Can the method distinguish unsynthesizable molecules from molecules where the search failed?
2. Why use Tanimoto similarity instead of canonicalized SMILES (with explicit stereochemistry/canonicalization rules)?
3. Could you systematically characterize how the 185 cases achieve score = 1 but have infeasible routes? See above for my lack of confidence in some routes that expert deemed False.
4. The data split appears very unconventional (107253/100/1). Why? This seems highly problematic?
5. Why does the paper only use beam size 5? Retrosynthesis benchmark frequently uses 50, and a reduction will artificially reduce route diversity/quality.

---

> ### Author Response · Authors · 2026-01-02
>
> **First of all, a more general point: all science stands on the shoulders of giants. In some sense, every paper builds on and tries to improve previous work. By that logic, any paper could be labeled “incremental” or “not novel enough,” but that’s not a helpful or accurate way to evaluate scientific progress. We are the first to perform an extensive validation—crucially including expert evaluation—of the round-trip score as a metric for molecular synthesizability. This bridges an important gap between data-driven molecule synthesis and practical drug design. In that respect, our contribution is genuinely novel.**
>
>
> **Weakness:** The scholarly quality falls below typical TMLR standards.
>
> **Rebuttal:** **That’s your perspective, but it doesn’t necessarily reflect the views of others. Several of our co-authors have worked in this field for many years, and we believe we’re well positioned to assess the value and significance of this work.**
>
> **Weakness1:** I believe the central methodological idea - round-trip accuracy - was explored rather extensively previously (e.g., Schwaller et al.), and the manuscript contains some mischaracterizations of related work.
>
> **Reuttal:** Context: Inspired by recent advancements that leverage forward reaction models (Sun et al., 2021) to enhance
> retrosynthesis algorithms and rank the top-k synthetic routes predicted by retrosynthetic planners (Schwaller et al., 2019b; Liu et al., 2024). Schwaller et al., 2019b appears to be the first work to propose round-trip accuracy, using a forward reaction model for evaluation. We believe this description is accurate.
>
>
> **Weakness3:** The scholarly quality falls below typical TMLR standards. While the concept is easy to follow, the presentation is confusing at times (e.g., “additional 501…” is not operationalized clearly enough for reproducibility). In other places, the manuscript spends disproportionate space on relatively standard definitions (e.g., nearly half a page on true/false positives/negatives).
>
> **Rebuttal:** We rewrote it to "For the 501 molecules whose predicted routes do not match the reference routes" and updated the paper.
>
>
> **Q1:** The round-trip metric is not novel. The paper repeatedly claims novelty for the “round-trip score” (e.g., Abstract: “a new, data-driven metric”; Section 1: “novel metric”), which I believe is factually incorrect. For example, Schwaller et al. (NeurIPS 2019 workshop) introduces round-trip accuracy, and related ideas also appear in Schwaller et al. (Chem. Sci. 2020) and LocalRetro (JACS Au 2021). A comprehensive work was also preprinted this year (Sadowski, arXiv:2510.10645). These papers may use different models and settings, but the manuscript cannot ignore prior efforts on round-trip-style validation.
>
> **Rebuttal:** The papers you cited use round-trip accuracy to evaluate single-step models. To our knowledge, Schwaller et al. (NeurIPS 2019 workshop) were the first to propose this metric. Beyond adopting the metric for evaluation, these later works do not appear to introduce methodological improvements to it. Therefore, requesting citations in this context is not quite appropriate.
>
> **Q2:** Relatedly, cited papers such as SynFlowNet/RGFN/SynCoGen do not merely use retrosynthesis planners to evaluate molecules; rather, they directly generate synthesizable pathways (bottom-up assembly) and then use retrosynthesis to check feasibility. In that sense, these works already incorporate a “generate + verify” approach, and they do so using more independent components than models trained on the same dataset. Similarly, RAScore/RetroGNN approximate synthesizability via proxies and then use retrosynthesis to check selected molecules, which are also forms of proxy/circular validation.
>
> **Rebuttal:** We carefully reviewed these papers and do not believe it is reasonable to rely on retrosynthesis models alone to evaluate synthesizability. As we state in the paper, the top-1 accuracy of retrosynthesis models is low, which makes search success rate an inherently unreliable metric. Retrosynthesis models only propose routes; therefore, additional verification metrics—such as exact-match checks or round-trip scores—are needed to validate whether the proposed solutions actually reproduce the target.

---

> ### Author Response · Authors · 2026-01-02
>
> **Weakness 2:** The paper frames the forward model as a “substitute for wet lab experiments,” but the forward model is trained on the same data distribution as the retrosynthesis planner. This approximation may lead to false agreement. In addition, there are further methodological concerns (see below).
>
> **Q3:** Both the retrosynthesis planner and the forward model are trained on similar USPTO-derived data. It is possible that a “successful” round-trip indicates the two models have learned the same biases or errors in the training data. This could explain why there is still a substantial false-positive rate (reported as ~24% in the manuscript). More generally, the current evaluation does not appear to robustly disentangle truly unsynthesizable molecules from cases where the search simply failed to find an available route. Lastly, I do not understand why Tanimoto similarity is needed here: if the goal is to check identity, canonicalized SMILES (with clear stereochemistry handling) seems simpler, and the current choice reads as an unnecessary complication.
>
> **Rebuttal:** 1. Although our round-trip score has a 24% false-positive rate, this is still 14 percentage points lower than the false-positive rate of search success rate (38%). This indicates that the round-trip score provides a more reliable evaluation metric.
>
> 2. Many molecules found in nature cannot be synthesized with current methods, while many man-made molecules are in fact synthesizable. Our goal is therefore not to classify molecules as “natural” vs. “synthetic,” but to use a round-trip score learned from available reaction data to identify molecules that are likely synthesizable. In other words, the round-trip score is a data-driven metric.
>
> 3. Since the round-trip score can serve as a reward signal to optimize the synthesizability of molecules generated by drug design models, we need a continuous-valued score rather than a binary 0/1 indicator.
>
>
> **Q4:** Lastly, a significant part of the work is due to manual validation of certain routes. However, I personally evaluated some routes that the experts found to be False (in supplemental information), and I deem MANY of them to be feasible. A short list of examples: Route 894 (amide coupling with chloroacetyl chloride and fluorobenzylpiperdinol should work, as secondary amines are a lot more nucleophilic compared to alcohols), Route 120 (Sonogashira coupling with a alkynoic acid and aryl halide, the carboxylic acid does not necessarily inhibit reactions), Route 407 (Williamson ether synthesis of a bromoether and a thiophene, appears very straightforward), Route 41 (carbamoylation of a secondary amine, seems quite possible), Route 21 (SnAr/N-arylation with a heteroaryl chloride and a secondary amide, this is a little tricky as amide nitrogens are weak nucleophiles, but I fully expect it to work with the right catalyst.)
>
> **Rebuttal:** Template-based retrosynthesis models often generate illusory or unrealistic synthetic routes, and relying solely on expert judgment can introduce bias. Therefore, we chose to evaluate route feasibility strictly using evidence from CAS SciFinder.
>
> **R1:** The authors must properly discuss all existing round-trip scores (Schwaller et al.) as well as related "circular-validation" works such as SynFlowNet. The paper needs to explain how this work differs from existing scores and compare against them. The novelty claims require significant revision, and the actual contribution must be clarified.
>
> **Rebuttal:** We have already stated in the paper that it is inappropriate to use a retrosynthesis model as the evaluation metric for search success rate. We have also removed the word “novel” from the manuscript.
>
>
> **R2:** The authors must address the fact that the planner and forward model are trained on the same dataset (USPTO). This should include a reaction-level split or ablation where the forward model is trained without reactions present in test routes (or without any reactions whose products match test targets), with re-reported Table 1/2-style results.
>
> **Rebuttal:** We believe there is a misunderstanding about the core of our method. Our goal is to learn a round-trip score for assessing molecule synthesizability using as much reaction data as possible. Naturally, more training data benefits both the retrosynthetic planner and the forward reaction model. If we already have reaction data to train the retrosynthetic planner, we can—and should—use the same data to train the forward model as well. Given this, we do not see a reason to reduce the amount of reaction data used to train the forward reaction model.

---

> > ### Author Response · Authors · 2026-01-02
> >
> > **R3** The authors should discuss known limitations of round-trip metrics (e.g., not verifying full chemical validity such as reagents/conditions/yields), as noted in prior work (e.g., Digital Discovery 2024, 558). This also requires toning down any claims that the method can substitute for “wet lab experiments.”
> >
> > **Rebuttal:** We added this limitation in the conclusion. We revised the claim.
> >
> > Less Critical:
> >
> > **1:** Can the method distinguish unsynthesizable molecules from cases where the search failed?
> >
> > No.
> >
> > **2:** Could you systematically characterize the 185 cases that achieve a score of 1 but still have infeasible routes? (Related to concerns about expert-labeled “False” routes.)
> >
> > These cases are primarily due to errors in the forward reaction model, which can occasionally assign a perfect round-trip score to an infeasible route.
> >
> > **3:** The data split seems very unconventional (107,253 / 100 / 1). Why? Isn’t this problematic?
> >
> > Our goal is to maximize the amount of data used to train the forward reaction model, so we allocate nearly all available reactions to training and keep only minimal data for validation and testing.
> >
> > **4:** Why does the paper only use beam size 5? Retrosynthesis benchmarks often use 50, and a smaller beam could reduce route diversity and quality.
> >
> > We use beam size 5 because prior work (e.g., FusionRetro) reports that beam size 5 is sufficient in practice.
> > Reference: FusionRetro: Molecule Representation Fusion via In-Context Learning for Retrosynthetic Planning, ICML 2023.

---

> ### Comment · Reviewer_ETi1 · 2026-01-02
>
> I reviewed the authors' rebuttal. While I recognize critical feedback can be frustrating, I find the rebuttal disappointing in both its tone and its scientific substance. Dismissing reviewer concerns with "that's your perspective" or appeals to co-author seniority does not constitute scientific engagement. The rebuttal reads unnecessarily adversarial and at points dismissive. My core scientific concerns remain largely unaddressed.
>
> 1. Inappropriate treatment of prior work. The core idea - round trip verification using a forward model - has clear precedents. Even if the present manuscript’s application differs (e.g. multi-step setting, which is a straightforward extension), proper scholarship requires explicitly crediting, __distinguishing, and comparing__ against foundational work. Dismissing the concern by saying “citations in this context is not quite appropriate” because they are "older" or "unimproved" is contrary to academic standards. I note the claim about 'multi-step' being novel is also not quite correct, as the Schwaller's Chem. Sci. 2020 already has round-trip validation within multi-step planning. Lastly, dismissing reviewer concerns regarding scholarly quality by stating that "Several of our co-authors have worked in this field for many years" is an appeal to authority that has no place in scientific discourse. Scientific validity is established by evidence, not by the seniority of the author list.
> 2. Circularity: This is perhaps the most significant methodological issue that the authors fail to address. When both the retrosynthetic planner and forward model are trained on the same USPTO data, a successful round-trip may indicate that both models learned the same biases rather than confirming true synthesizability. The authors' response ("Our goal is to learn a round-trip score... we do not see a reason to reduce the amount of reaction data") misses the point. I am not suggesting using less data, I am asking for ablation studies (e.g., reaction-level splits) to assess whether the metric's validity depends on this circularity. The refusal to engage with this methodological concern is disappointing.
> 3. In response to my query about the 107253/100/1 split, the authors admitted: "we allocate nearly all available reactions to training and keep only minimal data for validation and testing." If the test set is indeed a single data point (or even 100), the reported metrics are statistically meaningless. Establishing a new metric requires a large validation set.
> 4. Expert validation: I provided specific chemical reasoning for several routes. A response citing CAS SciFinder without engaging with the chemistry does not address my concern that the ground truth labels may contain systematic errors.
>
> I encourage the authors to re-read my original review and constructively address my comments in a significantly revised manuscript.

---

> > ### Author Response · Authors · 2026-01-02
> >
> > **1** Inappropriate treatment of prior work. The core idea - round trip verification using a forward model - has clear precedents. Even if the present manuscript’s application differs (e.g. multi-step setting, which is a straightforward extension), proper scholarship requires explicitly crediting, distinguishing, and comparing against foundational work. Dismissing the concern by saying “citations in this context is not quite appropriate” because they are "older" or "unimproved" is contrary to academic standards. I note the claim about 'multi-step' being novel is also not quite correct, as the Schwaller's Chem. Sci. 2020 already has round-trip validation within multi-step planning. Lastly, dismissing reviewer concerns regarding scholarly quality by stating that "Several of our co-authors have worked in this field for many years" is an appeal to authority that has no place in scientific discourse. Scientific validity is established by evidence, not by the seniority of the author list.
> >
> > **Rebuttal:** We have included the discussion in the introduction.
> >
> > **2** Circularity: This is perhaps the most significant methodological issue that the authors fail to address. When both the retrosynthetic planner and forward model are trained on the same USPTO data, a successful round-trip may indicate that both models learned the same biases rather than confirming true synthesizability. The authors' response ("Our goal is to learn a round-trip score... we do not see a reason to reduce the amount of reaction data") misses the point. I am not suggesting using less data, I am asking for ablation studies (e.g., reaction-level splits) to assess whether the metric's validity depends on this circularity. The refusal to engage with this methodological concern is disappointing.
> >
> > **Rebuttal:** There may be a misunderstanding here. In this work, it is reasonable to exploit the cyclical relationship between retrosynthesis and forward prediction. For example, CREBM[2] trains a forward reaction model on the same reaction data used to train the retrosynthesis model, and then uses that forward model to define an energy function for post-training. This energy-based re-ranking/post-training step improves the retrosynthetic planner’s route predictions by favoring routes that are more consistent under forward “execution.”
> >
> > [2] Preference optimization for molecule synthesis with conditional residual energy-based models, 2024 ICML.
> >
> > **3** In response to my query about the 107253/100/1 split, the authors admitted: "we allocate nearly all available reactions to training and keep only minimal data for validation and testing." If the test set is indeed a single data point (or even 100), the reported metrics are statistically meaningless. Establishing a new metric requires a large validation set.
> >
> > **Rebuttal**
> >
> > Please note that we use different dataset splits in Sections 4.1 and 4.2...
> >
> > **4** Expert validation: I provided specific chemical reasoning for several routes. A response citing CAS SciFinder without engaging with the chemistry does not address my concern that the ground truth labels may contain systematic errors.
> >
> > **Rebuttal** If you cannot provide wet-lab validation or cite relevant literature to support the claim, then you should not assert that your expert evaluation is correct.

---

### Decision · Action_Editor_WfH2 · 2026-03-14

**Recommendation:** Reject

**Audience:**

No

**Audience Explanation:**

While the topic of molecular synthesizability is relevant to the TMLR community, findings must be grounded in methodological rigor and scientific reliability. Given the unaddressed circularity issues, lack of comparison with SOTA baselines, and insufficient statistical foundation, the proposed "round-trip score" remains unvalidated.

**Claims And Evidence:**

No

**Claims Explanation:**

While this paper addresses an important problem in molecular synthesizability evaluation, all three reviewers recommend rejection based on fundamental flaws in methodology and the authors' failure to address critical concerns during the rebuttal. Main reasons are as follows:

The reviewers raised significant concerns regarding potential bias, as both the retrosynthesis and forward models were trained on highly similar data. The authors failed to provide the requested ablation studies to prove that the "round-trip score" reflects true chemistry rather than shared model artifacts.

The core concept has clear precedents (e.g., Schwaller et al., 2019), yet the manuscript claims high novelty without comparing the performance against existing round-trip implementations or SOTA methods like SynFlowNet.

The dataset construction is problematic, featuring an insufficiently small test set (1%) and manually annotated labels that lack cross-validation or inter-rater reliability metrics.

The authors failed to scientifically address the reviewers' critiques during the discussion phase. The rebuttal was perceived as dismissive and relied on "appeals to authority" rather than empirical evidence or technical clarification.

**Resubmission Of Major Revision:**

The authors may consider submitting a major revision at a later time.